REGISTERED REPORT PROTOCOL

# Evaluating equality in prescribing Novel Oral Anticoagulants (NOACs) in England: The protocol of a Bayesian small area analysis

Ehsan Rezaei-Darzi[1,2], Parinaz Mehdipour [1,3], Mariachiara Di Cesare[4], Farshad Farzadfar [1], Shadi Rahimzadeh[4], Lisa Nissen[5], Alireza Ahmadvand[5]*

**1** Non-Communicable Diseases Research Centre, Endocrinology and Metabolism Population Sciences Institute, Tehran University of Medical Sciences, Tehran, Iran, **2** Monash University Accident Research Centre, Monash University, Clayton, Victoria, Australia, **3** Centre for Epidemiology and Biostatistics, Melbourne School of Population and Global Health, University of Melbourne, Melbourne, Australia, **4** Department of Natural Sciences, School of Science and Technology, Middlesex University London, London, United Kingdom, **5** School of Medicine, Griffith University, Brisbane, Queensland, Australia

* a.ahmadvand@griffith.edu.au

This is a Registered Report and may have an associated publication; please check the article page on the journal site for any related articles.

## Abstract

### Background

Atrial fibrillation (AF) is the most common cardiac arrhythmia, affecting about 1.6% of the population in England. Novel oral anticoagulants (NOACs) are approved AF treatments that reduce stroke risk. In this study, we estimate the equality in individual NOAC prescriptions with high spatial resolution in Clinical Commissioning Groups (CCGs) across England from 2014 to 2019.

### Methods

A Bayesian spatio-temporal model will be used to estimate and predict the individual NOAC prescription trend on 'prescription data' as an indicator of health services utilisation, using a small area analysis methodology. The main dataset in this study is the "Practice Level Prescribing in England," which contains four individual NOACs prescribed by all registered GP practices in England. We will use the defined daily dose (DDD) equivalent methodology, as recommended by the World Health Organization (WHO), to compare across space and time. Four licensed NOACs datasets will be summed per 1,000 patients at the CCG-level over time. We will also adjust for CCG-level covariates, such as demographic data, Multiple Deprivation Index, and rural-urban classification. We aim to employ the extended BYM2 model (space-time model) using the RStan package.

### Discussion

This study suggests a new statistical modelling approach to link prescription and socioeconomic data to model pharmacoepidemiologic data. Quantifying space and time differences will allow for the evaluation of inequalities in the prescription of NOACs. The methodology will help develop geographically targeted public health interventions, campaigns, audits, or

**Data Availability Statement:** All relevant data from this study will be made available upon study completion.

**Funding:** The author(s) received no specific funding for this work.

**Competing interests:** The authors have declared that no competing interests exist.

guidelines to improve areas of low prescription. This approach can be used for other medications, especially those used for chronic diseases that must be monitored over time.

## Introduction

### Background

Atrial fibrillation (AF) is the most common cardiac arrhythmia [1]. Men are more often affected than women, and the prevalence of AF increases with age [2]. In the UK, the age-sex standardised prevalence of AF has increased from 2.14% in 2000 to 3.29% in 2016 [2]. AF is a major cause of ischaemic stroke, as the risk of stroke is five times higher than in a person with a normal heart rhythm [3]. The age-adjusted AF incidence and prevalence are lower in women; however, the absolute number of men and women living with AF remains similar due to men's shorter life expectancy [4].

The UK's National Institute for Health and Care Excellence (NICE) has approved four licensed non-Vitamin K antagonist oral anticoagulants (NOACs): Apixaban, Dabigatran, Edoxaban, and Rivaroxaban. Dabigatran is a direct thrombin inhibitor, while the others are direct factor Xa inhibitors. Large international multicentre trials have shown that NOACs reduce the risk of stroke in patients with non-valvular atrial fibrillation similar to warfarin. Their ease of use, minimal need for monitoring, and negligible interactions with other drugs have made NOACs a mainstream treatment choice among clinicians [5]. NOACs have limiting characteristics as well, such as the clinicians' inability to assess dosing, compliance, or wash out with an uncomplicated laboratory test, the lack of an antidote to rapidly control major haemorrhage, and reduced safety in emergent or urgent surgical procedures [6].

These NOACs are primarily prescribed to prevent stroke and systemic embolism in patients with non-valvular atrial fibrillation (AF) [7–10]. Additionally, they are indicated to treat deep vein thrombosis (DVT) and pulmonary embolism (PE), prevent and recurrent DVT and PE in adults, and to prevent atherothrombotic events after the management of acute coronary syndrome [11–15].

Although anticoagulation to reduce the risk of stroke is an essential part of managing AF, patients are not always appropriately anticoagulated [16]. In 2013, an estimated 7,000 strokes could have been avoided, and 2,100 lives saved each year in England with appropriate AF management [17]. The NICE Implementation Collaborative has identified barriers to NOACs use at general practice levels, which include, but are not limited to:

- the continued use of aspirin for stroke prevention,

- health care professionals' concerns regarding patient adherence, as there is no need for routine coagulation monitoring with NOACs,

- the cost of prescribing NOACs in comparison to alternatives such as vitamin K antagonists (VKA), like Warfarin, or

- the unavailability of specific antidotes for NOACs (except for Dabigatran, for which idarucizumab is available) to reverse the drugs' effect in the event of a major bleed.

The NICE Implementation Collaborative explains that primary care providers prescribing NOACs need local leadership. Not all GPs can be expected to be experts in anticoagulation for atrial fibrillation. As the prevalence of atrial fibrillation continues to increase with age, local anticoagulant "champions" will be needed [3]. Also, in terms of the local care pathways, NICE-

approved treatments must be made available for prescribing NOACs. The NHS's Clinical Commissioning Groups (CCGs) have flexibility in making this happen and different models can be used to suit local needs.

The above-mentioned barriers to prescribing NOACs, plus variations in local leadership, general practitioners' expertise and confidence in prescribing NOACs, and CCG flexibility in prescribing cause variations across CCGs and over time. An AF diagnosis and anticoagulant-prescribing performance have practice-level variations [2]. There will be regional variation in stages of transiting to optimised NOAC prescriptions, with some relation to CCG-level characteristics in either burden of the clinical indications (AF) or barriers to switching to NOACs. Currently, little is known about the prescribing patterns at smaller geographical and administrative levels, e.g. CCGs, which can be a valuable index for economic and health service planning.

## Aims and questions

NHS Digital has made large data and geographical information about prescription patterns across NHS available. This has made providing estimates of prescription patterns at smaller geographic levels possible and will help inform decisions and policymaking.

We aim to develop and initiate an analytical strategy for small-area estimates. This research will integrate concepts and methods from the fields of medicine, clinical epidemiology, population health, and statistics to provide evidence for policy and programmatic decisions regarding the prescribing patterns of NOACs in local populations. The objective of this research is to quantify–for the first time for individual NOACs–the prescribing patterns at "small-area" CCG levels. This will subsequently adjust the spatio-temporal prescribing patterns according to relevant covariates across CCGs in England (space) and over time.

This higher resolution information about NOACs across CCGs can demonstrate major differences in their prescribing patterns, potential variables that have contributed to these differences, and the amount of deviation. This information will allow policymakers to deliver feasible and cost-effective primary care interventions to improve and optimise NOAC prescribing at the population level.

## Methods

This study explains the statistical methodology for estimating and predicting 'prescribing data' as an indicator of health services utilisation, using a small area analysis methodology.

The statistical methods in this study 'borrow strength' over time and space in a Bayesian framework. We will start from individual NOACs and correlate them over space (e.g., using a conditional autoregressive model) across CCGs, and over time (e.g., using random walk). The study will focus on four licensed, guideline-approved, available NOACs across the NHS. The international non-proprietary names (INNs) [generic names] of the NOACs are Apixaban, Dabigatran, Edoxaban, and Rivaroxaban.

### The rationale for analysing NOACs

Our study will analyse prescribing patterns for NOACs according to the following:

1. Clinical: NOACs are prescribed for clinically significant, diverse, and priority indications, i.e., AF, prevention of DVT and PE, or after acute MI. The licensed NOACs can be prescribed by general practitioners in England. Also, there is scientific evidence suggesting that the NOACs are not being prescribed optimally, and interventions are needed to increase

their use in primary care [18]. However, the prescribing patterns between CCGs remain unknown.

2. <u>Health economics</u>: NOACs are costly medications covered by the NHS. According to the NHS's Prescribing by GP Practice datasets, Apixaban and Rivaroxaban had the top two highest ACT and NIC costs in July of 2019.

3. <u>Technical statistical advantage</u>: for small area modelling using spatio-temporal analysis, the above mentioned four NOACs have a unique advantage. Prescribing one is mutually exclusive in prescribing the others, which means that they cannot be co-prescribed. Therefore, each medication can be modelled and interpreted individually.

## Identifiers of individual NOACs

This study focuses on four individual NOACs with unique identifiers in the intended prescribing datasets, including their corresponding British National Formulary (BNF) codes. The BNF codes show what medication has been prescribed. Additionally, for aggregating the different dosage forms of these four NOACs, we will add the individual alphanumeric codes developed by the World Health Organisation (WHO), i.e., the Anatomical Therapeutic Chemical (ATC) Classification System. ATC classifies the active ingredients of drugs according to the organ or system on which they act and their therapeutic, pharmacological, and chemical properties.

## Standardising different dosages and primary variables of interest

We will use the defined daily dose (DDD) methodology, as recommended by WHO, to provide a fixed unit of measurement independent of price, currency, package size, and strength. This enables us to assess trends in medication prescribing patterns and to compare different geographical areas over time. By definition, "DDD is the assumed average maintenance daily dose for a drug administered for its main indication in adults." Table 1 summarises the individual NOACs' identifiers, DDDs, and available dosage forms in the UK.

As a practical example, if 100 Edoxaban at 30 mg and 200 Edoxaban at 60 mg are prescribed at any point, the total DDD equivalent for Edoxaban is calculated as:

$$(100 \text{ x } 30 \text{ mg} + 200 \text{ x } 60 \text{ mg})/60 \text{ mg} = 250$$

**Table 1. Identifiers of the individual NOACs.**

| Name | ATC code | BNF Code | DDD* (mg) | Dosage Forms |
|------|----------|----------|-----------|--------------|
| Apixaban | B01AF02 | 0208020Z0 | 10 | 2.5mg |
| | | | | 5mg |
| Dabigatran etexilate | B01AE07 | 0208020X0 | 300 | 75mg |
| | | | | 110mg |
| | | | | 150mg |
| Edoxaban | B01AF03 | 0208020AA | 60 | 30mg |
| | | | | 60mg |
| Rivaroxaban | B01AF01 | 0208020Y0 | 20 | 2.5mg |
| | | | | 10mg |
| | | | | 15mg |
| | | | | 20mg |

* DDD: Defined daily dose.

This total 'DDD equivalent' is unit-free and will serve as a generic parameter that can be compared at various locations over time.

However, to compare CCGs with each other and any CCG in different years regarding the total DDD equivalent, population changes over space and time must be considered. Therefore, we will divide the calculated total DDD equivalent by the population and then multiply the result by 1,000. This will give a continuous outcome variable, in the form of a rate called DDD per 1,000 population, which will be the primary variable in the analyses.

### Adjusting for covariates

To account for variables that may explain the distribution of the primary variable of interest, we will use population-level data to include relevant covariates in the analytical model, including, but not limited to, age, socio-economic indicators, the number of prescribing practitioners, and the prevalence of specific medical conditions, such as atrial fibrillation.

### Visualising outputs

To visualise the analysis outputs, we will use the Clinical Commissioning Groups (CCG) Boundaries as of April 2019, available from the Office for National Statistics website under the Open Government Licence v3.0 [19,20]. In compliance with copyright, we initially downloaded and reproduced a CCG map shapefile using R software version 3.5.1 (Fig 1).

### Reference year

For meaningful comparison between CCGs over time, we will set the 2019 calendar year as the reference time and will consider providing retrospective comparisons back to 2014.

### Data sources

The main dataset in this study will be the "Practice Level Prescribing in England," a list of all medicines, dressings, and appliances that are prescribed by all registered GP practices in England [21].

Practice Level Prescribing in England is available from August of 2010, and updated monthly, covering the specifics of each item prescribed. The data covers England NHS's prescriptions and dispensation in the UK. Prescriptions that are written in England but dispensed outside of England are also included. The data includes prescriptions written by GPs and other non-medical prescribers, such as nurses and pharmacists attached to GP practices. Medications are identified by their British National Formulary (BNF) code. The practices listed include all those registered in England and several "dummy" practices created by Primary Care Trusts (PCTs) to identify prescriptions in certain environments or circumstances, including specialist clinics, hospices, prisons, and training units.

Each monthly data set is over 10 million rows. The data includes the total quantity of individual treatments prescribed for each practice identified by the BNF code. Six-calendar years of Apixaban, Rivaroxaban, Edoxaban, and Dabigatran etexilate prescriptions are extracted from January of 2014 to December of 2019 to form the main dataset. Databases used in this study do not contain clinical diagnoses. The study does not aim to differentiate between different indications for prescribing NOACs.

The GP practice list size (the number of registered patients) in five-years age bands and a sex distribution is available quarterly from January of 2014 and monthly from April of 2017 [22]. A linear interpolation will be conducted to cover unsupported demographic data, assuming a linear change in the sex and age discrepancy [23]. To capture the deprivation in England,

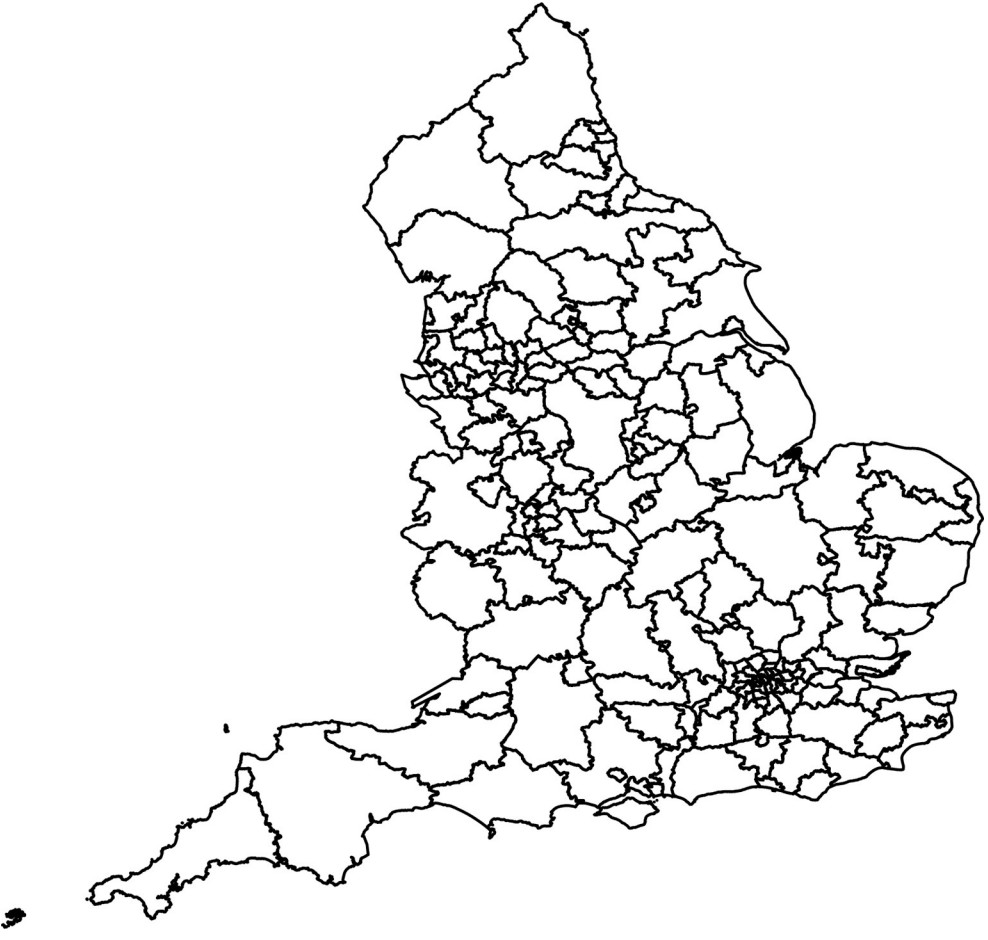

**Fig 1. Reproduced CCG map shapefile for England as of April 2019.** Source: Office for National Statistics licensed under the Open Government Licence v.3.0. Contains OS data © Crown copyright and database right [2020].

the Index of Multiple Deprivation (IMD) 2019 by CCG is extracted from the Ministry of Housing, Communities & Local Government (MHCLG) [24,25]. The Overall Index of Multiple Deprivation is produced according to the seven domains of deprivation, with a particular weights approach (income (22.5%); employment (22.5%); education, skills and training deprivation (13.5%); health and disability (13.5%); crime (9.3%); barriers to housing and services (9.3%); living environment (9.0%)). The 2011 rural-urban classification (RUC) data by CCG is obtained from the Office for National Statistics (ONS), including population data [26].

## Data processing

GP practice data were summed on the total quantity of each defined BNF code per CCG over time. The aggregated main data were linked to the demographic data to match standardised DDDs for individual NOAC identifiers per 1,000 patients for the average age and proportion of males. The CCG-level IMD summarised score for 2019 and RUC data from 2011 were merged regardless of their time effects. According to the boundary and name changes during the years, some of the CCGs had been updated or were merged [27]. Our data frame consisting of 191 CCGs, over six study years, was a total of 1146 records for standardised DDD of individual NOACs.

## Analysis and modelling

Small-area health studies have two main features: the spatial location and the distribution of disease, which is known as georeferenced disease data [28]. It is important to use a proper analysis method [28]. Spatial models help quantify inequalities in drug prescriptions and assess trends. The model's estimates for each district depend on its data and neighbours' information [29]. Due to insufficient sample size at small geographical levels, direct estimators are too unreliable to provide adequate estimation, while Bayesian methods have improved estimates [30]. Bayesian methods are commonly used in modern statistical packages to facilitate quick computational algorithms, which was not feasible in the past [31].

The small area estimation method using the Bayesian technique can help count rare events in regions with a small population. In the Bayesian approach, we fit data to the model structure, add information, and perform an external validation. The Bayesian methods enable researchers a sensible interpretation from the statistical concepts, directly quantifying uncertainty and incorporating complex issues [32]. Hierarchical Bayesian modelling is used to estimate the incidence or prevalence rate in spatial epidemiology. In Bayesian hierarchical models, parameters have distributions based on prior beliefs defined by expert opinions or study investigators. These distributions control the parameter limits, which can vary in the model.

When bordering zones show a higher correlation than remote zones, we have real data with a spatial structure [33]. Hence, Besag's intrinsic conditional autoregressive (ICAR) model can account for the spatial autocorrelation by putting information in from adjacent areas [34]. Although this model smooths the noisy estimates, it couldn't explain the variability of these data entirely. The BYM model was introduced by Besag et al. in 1991 to combine both spatial and non-spatial random effects to account for all data variation [35]. However, it is difficult to assume independence between these two components in the BYM model. Nearly all variation can be addressed since the non-spatial random effect is included to capture the independent region-specific variation. Consequently, it is not possible to split the variability over the effects. The BYM2 model, a reparameterization of the BYM model, addresses this issue to interpret parameters and select hyper-priors for spatial and non-spatial precision [36]. The new model modified the variability distribution between two components using a single-precision parameter for the combined component and a mixing parameter for the amount of spatial to non-spatial variation [36]. To quantify the temporal trends in the data, the spatial model can be extended to a space-time model by adding a temporal term in the small areas [37]. We plan to employ the extended BYM2 model (space-time model) using an RStan package for GP Practice Presentation-level Data.

## Small-area estimation model

The statistical analysis will be carried out using a Bayesian hierarchical framework. We will be able to investigate the geographical distribution of the outcome of interest, DDDs quantity. Let $y_i$ denote the DDDs quantity in each area (i). Since DDDs are a rare count measure, a Poisson distribution is usually used. In this case, areas with low DDDs quantity frequency might have small expected numbers, and sampling variability will occur with large variance. Due to this potential, Bayesian hierarchical models are used to achieve spatial smoothness of estimates [28].

A Poisson model will be used to map DDDs spatial distribution over time and to explore other related factors associated with DDDs at the CCG-level. The effects of clustering within the CCGs (i.e., patients travelling between different clinics but within a specific assigned CCG) will be present in the remaining error terms.

The DDDs quantity of individual NOACs is represented by $y_{ij}$ in region i and time j. Let $E_{ij}$ denote the expected number of relevant people in area i and time j, which can be calculated based on the demographic data set. A Poisson distribution is used to model the DDDs quantity, given $\theta_{ij}$. It denotes the underlying true time and area-specific relative rate. The estimation of $\theta_{ij}$ is $\frac{y_{ij}}{E_{ij}}$, which corresponds to timely rates.

$$y_{ij}|\theta_{ij} \sim \text{Poisson}(E_{ij} \times \theta_{ij})$$

A general model formulation assumes that the log relative rate $\mu_{ij} = \log(\theta_{ij})$ has a decomposition as below.

$$\mu_{ij} = \beta_0 + \beta x_{ij}^T + \kappa_i * \boldsymbol{\sigma} + \beta_3 time_{ij}$$

The lognormal Poisson model includes both spatial smoothing and a random effect for non-spatial heterogeneity. The DDDs quantity ($\mu_{ij}$) in area i and year j measures the space and time variations in the data. The spatial structure of the data, according to the BYM2 as described by Riebler et al., 2016, includes an overall DDDs yearly rate at the country level ($\beta_0$), CCG-level covariates ($\beta x_{ij}^T$), a combined random effects component ($\kappa_i$) consisting of both spatial and non-spatial random-effects, and the temporal effect. This mixture of components consists of either spatial and non-spatial random effects to account for model error terms. The non-spatial error term is used to consider over-dispersion not modelled by the Poisson variates. In the latest version of the BYM models, a combination of these two components is incorporated in the Poisson model to make it more interpretable and allow for sensible hyperparameters [36].

$$\kappa_i = \eta_i((\sqrt{\boldsymbol{\rho}}/\text{s}) + v_i((\sqrt{1-\boldsymbol{\rho}}))$$

Information between adjacent areas can be captured with spatially-correlated random effect ($\eta_i$), which allows for sharing the similarity of characteristics. Spatially uncorrelated random effect accounts for heterogeneity within areas ($v_i$). The mixture component $\kappa_i$ smooths observations to the total mean $\mu_{ij}$, with a precision parameter $\sigma$ (overall standard deviation for combined error terms) and weighting parameter $\rho$ for spatial/non-spatial variation. $\rho$ has a value between zero and one and models the amount of the variance that comes from the spatially correlated error terms over the variance that comes from the independent error terms. In this formula, s is the scaling factor that scales the proportion of variance $\rho$ and lets $\sigma$ be the standard deviation of the combined components. Spatial variation due to different amounts of DDDs in each CCG and geographical inequalities in CCGs is captured by this combined parameter. The temporal component $\beta_3$ smooths variation over time (yearly) and considers potential time correlation. We will account for exposure time by including it as a variable in the Poisson model. However, in the future, we will consider other time models that could fit our data.

We will develop the inferencing of the Bayesian model in the open-source RStan package, which is a highly expressive general probabilistic programming language for the specification of Bayesian statistical models [38–40]. Stan used No-U-Turn Sampler, or NUTS, an extension of the Hamilton Monte Carlo algorithm sampler, to draw samples from the model parameters and residual errors from the posterior distribution. This algorithm, introduced by Hoffman et al. in 2014, efficiently minimizes manual interventions and allows users to save time and focus on model development [41]. NUTS is a simpler algorithm that is used to select sample points that have a wider distribution to prevent redundant sampling steps [41]. For hierarchical models comprising a complex posterior, such as the BYM models, Stan's NUTS sampler makes more robust estimates compared to the Gibbs or Metropolis samplers [33].

The posterior summaries, including the median and 95% credible interval (the 2.5th and 97.5<sup>th</sup> percentiles) for each parameter, will be calculated from the drawn samples [41].

## Assumptions for the data and the model

The outputs of our model may be different from the actual population characteristics because of sampling and non-sampling errors in the data and assumptions underlying the modelling techniques. In the modelling and analysis, we may need to make a few assumptions, such as none of these four NOACs are withdrawn from commercial markets, prescribing patterns or authorisation for these NOACs do not change significantly over time (for example, via a major change in clinical practice guidelines), our estimates will also be model-unbiased under the assumption of the linear association between the response variable and the area-specific covariates when only area-specific auxiliary information is available, and all small (CCGs) and large areas (country) have the same characteristics [42].

These assumptions are made because they have implications in interpreting model output. If any of the assumptions appear to be true, the effects on the model's output will be checked through empirical-defining counterfactual scenarios (*a posteriori)* and rerunning the model.

## Model validation

To assess the validity of our estimations, we will conduct a sensitivity analysis in two stages [43]. In the first stage, we will randomly mask 10% of our data points and we will repeat all of the models for the remaining 90% of the data. We will use the (average) root mean squared error (RMSE) as a measure for the average squared difference between model estimates and the observed values. The RMSE is often used to measure the differences between the values predicted by a model and the observed values, a useful measure to capture model precision [44]. In the next stage, we will calculate the proportion of data points in our masked data set that fell within the 95% uncertainty interval of the withheld data.

The validation framework will check each model's performance using the summary of the parameters and their various quantities. These include the posterior mean, the posterior standard deviation, and various quantiles computed from the draws. We will check MCMC model-fitting measurements, including the Monte Carlo standard error (se_mean), the effective sample size (n_eff), and the R-hat statistic (Rhat).

## Ethical considerations

This study uses publicly-available data only, so no ethical approval is required. NHS and ONS data sets have open government licenses and we will cite the principal investigators of the secondary data sets.

## Open data sharing

The Research Data Australia platform will be used to make the models, codes, and detailed outputs available to the public and professionals [45]. Research Data Australia, an Australian Government-supported data discovery service of the Australian Research Data Commons, helps in finding, accessing, and reusing research data. Data will be stored on authors' university-affiliated storage platforms, with descriptions of and links to the data provided on Research Data Australia.

## Discussion

To our knowledge, this study is the first small-area analysis of the distribution of NOACs in England using the Bayesian approach. In comparison to warfarin, the vitamin K antagonists Dabigatran, Apixaban, Rivaroxaban, and Edoxaban have proven to be comparably effective in preventing stroke in AF and in treating venous thromboembolism. They are associated with a reduced risk of intracranial bleeding. NOACs' superiority compared to vitamin-K anticoagulants has been also acknowledged by the WHO, who has included Dabigatran (as representative of the pharmacological class) on the 21st WHO Essential Medicine List [46].

However, as the selection of a particular NOAC will depend on a few factors, the prescribing patterns are different across geographic areas and over time. These selection factors can be individual-related factors, such as renal function, possible drug-drug interactions, or preferred dosing schedules (once- or twice-daily), prescriber-related factors, such as familiarity with NOACs and their dosing or being comfortable prescribing NOACs; and system-related factors, such as the availability of individual NOACs at a particular CCG.

### Research implications

This study will identify possible similarities or differences in prescribing individual NOACs over time and space to help identify possible gaps in NOAC prescriptions at the CCG level. Specifically, quantifying spatio-temporal differences will enable the evaluation of inequalities in prescribing NOACs. This quantification will be meaningful in developing geographically targeted public health interventions, campaigns, audits, or guidelines to improve low-prescribing areas.

Moreover, the spatio-temporal analysis in this study will be the fundamental framework for visualising variations in prescribing NOACs over time and highlighting possible geographical clusters, or 'hot-spots', of NOAC prescriptions.

The Bayesian spatio-temporal modelling of prescribing patterns for NOACs will help predict future patterns and provide estimation based on hypothetical scenarios or sensitivity analyses. It will assess counterfactual prescription scenarios for better prescriber preparedness outcomes at the CCG level and support decision making at the prescriber or CCG level.

### Target audiences of this research

Public health researchers, individual clinician prescribers, such as general practitioners and nurse practitioners, CCG managers across England, pharmacists working with GP practices, and NHS Implementation Collaborative groups are the target audiences of this research.

### Pros of this analytical approach

This study emphasizes the intersection of time and space in pharmacoepidemiology studies. A small number of studies consider the effects of combining time and location to estimate drug trends. Some researchers map the distribution of prescribing geo-referenced data by applying a likelihood method to a specific time. This paper is the first to use a hierarchical Bayesian spatiotemporal model to estimate standardised drug quantities of prescription data in small areas. The Bayesian hierarchical framework is more flexible and handles small amounts of data with spatial correlation. Bayesian hierarchical methods enable smoothing by borrowing information from neighbouring units, which leads to more stable estimates. Using an RStan package is another advantage of this study. This powerful programming language allows for disconnected subgraphs and island regions and better estimates of models with complex posteriors, such as the BYM model [33].

## Future directions

The future direction of this research includes the economic evaluation of prescribing NOACs at the small-area level, dynamic or real-time visualisation of analytical outputs for NOACs to translate our findings into practice and policy, predicting future patterns, and conducting this small-area analysis for other medication classes (individually or in groups). Additionally, the analytical approach from this study can be used for more detailed comparisons with other countries, including Canada, Australia, or New Zealand, depending on data availability.

## Considerations and limitations of the study

This study uses aggregate, population-level data, which is a similar design approach to ecological studies, without any gender- or age-specific data. Therefore, interpreting the results and outputs of this spatio-temporal small area analysis should be done at the CCG level, not at individual prescriber or patient levels, to prevent any ecological fallacies.

The main dataset that we will use for the Bayesian analysis, although rich in medication-related information, contains information for other relevant variables or covariates. Therefore, the inclusion of other covariates for statistical adjustment purposes depends on their availability from reliable sources over time.

Additionally, the main dataset that we will use for analysis will only cover CCGs in England, not the United Kingdom. Therefore, geographical interpretation and visualisation of the outputs may be limited.

The Prescribing by GP Practice database covers prescriptions by general and nurse practitioners. Data from other authorised prescribers, such as specialists or trainees working at hospitals or private practices, are not reflected in this dataset or analysis. General practitioners on average prescribe approximately 60–65% of all medications across NHS. However, the actual percentage of NOACs prescribed by GPs is unknown.

The four licensed NOACs analysed in this study are known by their generic names in the main data set. Therefore, parties interested in specific brand-focused information need more details, such as the market share of a brand, to translate the outputs of this study to their practice.

For optimal and meaningful interpretation, statistical model assumptions should be considered. Otherwise, reliable interpretation may not be possible. Finally, this study requires very large data sets, which makes replicating the methodology more suitable for advanced statistical software or packages.

## Conclusion

This study offers a new statistical approach to modelling pharmacoepidemiologic data. The generic analytical approach of this study can be applied to other medications, especially those prescribed for chronic conditions that must be taken for a long time (possibly a lifetime).

## Author Contributions

**Conceptualization:** Alireza Ahmadvand.

**Data curation:** Ehsan Rezaei-Darzi.

**Methodology:** Ehsan Rezaei-Darzi, Parinaz Mehdipour, Alireza Ahmadvand.

**Supervision:** Alireza Ahmadvand.

**Validation:** Ehsan Rezaei-Darzi.

**Visualization:** Ehsan Rezaei-Darzi.

**Writing – original draft:** Ehsan Rezaei-Darzi, Parinaz Mehdipour, Alireza Ahmadvand.

**Writing – review & editing:** Ehsan Rezaei-Darzi, Parinaz Mehdipour, Mariachiara Di Cesare, Farshad Farzadfar, Shadi Rahimzadeh, Lisa Nissen, Alireza Ahmadvand.

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
