## [Decision Letter · Decision Letter 0]

27 Aug 2020

PONE-D-20-10082

Evaluating Equality in Prescribing Novel Oral Anticoagulants (NOACs) in England: Protocol of a Bayesian Small Area Analysis

PLOS ONE

Dear Dr. Ahmadvand,

Thank you for submitting your manuscript to PLOS ONE. After careful consideration, we feel that it has merit but does not fully meet PLOS ONE’s publication criteria as it currently stands. Therefore, we invite you to submit a revised version of the manuscript that addresses the points raised during the review process.

Reviewer #1 has made some relevant remarks. In particular, I recommend the authors improve the clarity of their writing in the Aims and questions section. 

We look forward to receiving your revised manuscript.

Kind regards,

Michele Tizzoni

Academic Editor

PLOS ONE

Journal Requirements:

2.We suggest you thoroughly copyedit your manuscript for language usage, spelling, and grammar. If you do not know anyone who can help you do this, you may wish to consider employing a professional scientific editing service.  

3.Thank you for stating the following financial disclosure:

 [The funders had and will not have a role in study design, data collection and analysis, decision to publish, or preparation of the manuscript.].

5.We note that [Figure(s) 1] in your submission contain [map/satellite] images which may be copyrighted. All PLOS content is published under the Creative Commons Attribution License (CC BY 4.0), which means that the manuscript, images, and Supporting Information files will be freely available online, and any third party is permitted to access, download, copy, distribute, and use these materials in any way, even commercially, with proper attribution. For these reasons, we cannot publish previously copyrighted maps or satellite images created using proprietary data, such as Google software (Google Maps, Street View, and Earth). For more information, see our copyright guidelines: http://journals.plos.org/plosone/s/licenses-and-copyright.

1.    You may seek permission from the original copyright holder of Figure(s) [1] to publish the content specifically under the CC BY 4.0 license. 

Additional Editor Comments (if provided):

I apologies for the long time needed to complete the assessment but the COVID-19 pandemic has made very hard to find independent referees available for review.

Reviewers' comments:

Reviewer's Responses to Questions

**Comments to the Author**

1. Does the manuscript provide a valid rationale for the proposed study, with clearly identified and justified research questions?

Reviewer #1: Yes

2. Is the protocol technically sound and planned in a manner that will lead to a meaningful outcome and allow testing the stated hypotheses?

Reviewer #1: Yes

3. Is the methodology feasible and described in sufficient detail to allow the work to be replicable?

Reviewer #1: Yes

4. Have the authors described where all data underlying the findings will be made available when the study is complete?

Reviewer #1: No

5. Is the manuscript presented in an intelligible fashion and written in standard English?

Reviewer #1: Yes

6. Review Comments to the Author

You may also provide optional suggestions and comments to authors that they might find helpful in planning their study.

Reviewer #1: The paper describes a protocol for a study that will estimate the equality in prescribing individual NOACs with high spatial resolution in Clinical Commissioning Groups (CCGs) across England from 2013 to 2019, using Bayesian spatio-temporal modelling.

The introduction is focussed on the positive aspects of NOACs and does not refer to the negative or adverse effects associated with these medicines. A balanced review would be useful.

Placement of references should be at the end of a statement/sentence not before e.g. ‘These NOACS… for (5-8) 1)…’ it is confusing to have the references and then numbering of points.

Page 4 – ‘epidemic of atrial fibrillation’ is used but ‘epidemic’ is probably not the right word to use here.

Aims and questions section appears very long. The first sentence requires re-wording – it may not be required since it is covered already.

Methods – is the prescribing data linked to diagnosis? Will other indications other than Afib be considered in the analysis?

A reference to the ‘non-optimal’ prescribing is required.

How is clustering within areas accounted for?

2019 is used as a reference year, but the data are available april 2013-april 2019, so is it the first 4 months of 2019 is reference or the year May2018-April 2019? Not clear

How far forward will be considered for predictions?

Could age/gender specific DDDs be considered as large differences in age and gender specific prescribing?

Analysis and modelling section – more explanation of the Bayesian models would be included and why they are used over non-Bayesian approaches? Also, a brief intro to Hierarchical modelling – not all readers may be familiar.

Statement ‘BYM introduced by (30)….’ Please give authors name etc and not a reference. Also how does BYM address the issue with inter-dependency? Some explanation would be helpful

How was the temporal term defined? Monthly, yearly??

SAE model – provide references or explain the Bayesian Hierarchical framework. This section could be much clearer – particularly when explaining ‘spatial variation due to different…. By this combined parameters’. This was not clear.

‘NOVAs’ instead of ‘NOCAs’ needs correcting.

Explain why using log normal used?

Formula is used without explanation of the meaning of the parameters, It is assumed the reader is aware of these.

For the explanation of Rstan ‘Stan used no … from posterior’ requires further referencing. Why does NUTs sampler make ‘appropriate estimate’ over others – the justification is not evident.

Assumptions – how will these assumptions be tested? Should sensitivity analysis be applied here?

Validation – presented in the ‘past’ tense which reads as if this work/analysis has been done already, even though this is a protocol for the analysis. Please correct.

Ethical considerations – similarly using ‘past’ tense. Is this appropriate?

Discussion – there is some repetition with what is included in the introduction - is this required in a protocol?

There is no mention of whether the data and/or models will be made available through open source platforms. This is required.

The level of English could be improved e.g. pg 2 discussion last sentence ‘ can be monitor for a long time’ rather than ‘monitored’. A thorough proof-read of the paper is required.

7. PLOS authors have the option to publish the peer review history of their article (what does this mean?). If published, this will include your full peer review and any attached files.

Reviewer #1: No

---

## [Author Response · Author response to Decision Letter 0]

7 Oct 2020

Thanks for the time and valuable comments. We have uploaded our responses to editor and reviewer in the "Response to reviewers" file.

---

## [Decision Letter · Decision Letter 1]

16 Nov 2020

PONE-D-20-10082R1

Evaluating Equality in Prescribing Novel Oral Anticoagulants (NOACs) in England: Protocol of a Bayesian Small Area Analysis

PLOS ONE

Dear Dr. Ahmadvand,

Thank you for submitting your manuscript to PLOS ONE. After careful consideration, we feel that it has merit but does not fully meet PLOS ONE’s publication criteria as it currently stands. Therefore, we invite you to submit a revised version of the manuscript that addresses the points raised during the review process.

The Reviewer has identified some minor issues that should be taken into account in a revised version of the manuscript.

We look forward to receiving your revised manuscript.

Kind regards,

Michele Tizzoni

Academic Editor

PLOS ONE

Reviewers' comments:

Reviewer's Responses to Questions

**Comments to the Author**

1. Does the manuscript provide a valid rationale for the proposed study, with clearly identified and justified research questions?

Reviewer #1: Yes

2. Is the protocol technically sound and planned in a manner that will lead to a meaningful outcome and allow testing the stated hypotheses?

Reviewer #1: Yes

3. Is the methodology feasible and described in sufficient detail to allow the work to be replicable?

Reviewer #1: Yes

4. Have the authors described where all data underlying the findings will be made available when the study is complete?

Reviewer #1: Yes

5. Is the manuscript presented in an intelligible fashion and written in standard English?

Reviewer #1: No

6. Review Comments to the Author

You may also provide optional suggestions and comments to authors that they might find helpful in planning their study.

Reviewer #1: Most of the previous points have been addressed - however, there are two further minor corrections required.

1. Original Question in first review: Could age/gender specific DDDs be considered, as large differences in age and gender specific prescribing?

The response and change to the manuscript text is not what was requested. I understand that the data are not available by age or gender, and therefore, the following sentence in methods can be removed as it adds confusion rather than clarity.

‘DDD is not age- or gender-specific, and therefore, it can suitably be used in our modelling which is on

population-level, aggregate prescription data without gender or age specifications.’

2. Thorough Proof-read: Despite a request for a thorough proof-read for English, some errors still need to be addressed. A list of examples is provided, but not exhaustive:

In the methods

‘Health economical ‘– change to health economics

‘In compliance with the copy rights,’ – change to ‘in compliance with copyright’

‘Due to insufficient sample size prescribed at the small levels, direct estimators are unreliable to provide adequate..’ - please rephrase ‘at the small levels’ so it is clear what this means?

‘In Bayesian approach’ – should be ‘In the Bayesian approach..’

‘These distributions control the parameters limits that they can vary in the model.’ – rephrase

‘Since the non-spatial random effect is included to capture for independent region-specific variation, most or all of the variation can be addressed.’ – rephrase e.g. ‘Since the non-spatial random effect is included to enable the capture of the independent region-specific variation,….’

‘precision parameter for combined component and a mixing parameter..’ – change to ‘precision parameter for the combined component and a mixing parameter’

‘The effects of clustering within the CCGs (….) will show itself in the remainder error terms of the model.’ – change to ‘the effect of clustering with the CCGs () will be present in the remaining error terms…’

‘If any of the assumptions seems to be true, the main way … through empirical defining counterfactual..’ – change to ‘ If any of the assumptions appear to be true, the main way … through empirically defining counterfactual’

7. PLOS authors have the option to publish the peer review history of their article (what does this mean?). If published, this will include your full peer review and any attached files.

Reviewer #1: No

---

## [Author Response · Author response to Decision Letter 1]

6 Jan 2021

In response to editor and reviewer, we have revised and improved the language of the paper using a professional English language editting system. We have also attached a document "Response to Reviewer". 

Thank you.

---

## [Decision Letter · Decision Letter 2]

19 Jan 2021

Evaluating Equality in Prescribing Novel Oral Anticoagulants (NOACs) in England: The Protocol of a Bayesian Small Area Analysis

PONE-D-20-10082R2

Dear Dr. Ahmadvand,

We’re pleased to inform you that your manuscript has been judged scientifically suitable for publication and will be formally accepted for publication once it meets all outstanding technical requirements.

Kind regards,

Michele Tizzoni

Academic Editor

PLOS ONE

Additional Editor Comments (optional):

Reviewers' comments:

Reviewer's Responses to Questions

**Comments to the Author**

1. Does the manuscript provide a valid rationale for the proposed study, with clearly identified and justified research questions?

Reviewer #1: Yes

2. Is the protocol technically sound and planned in a manner that will lead to a meaningful outcome and allow testing the stated hypotheses?

Reviewer #1: Yes

3. Is the methodology feasible and described in sufficient detail to allow the work to be replicable?

Reviewer #1: Yes

4. Have the authors described where all data underlying the findings will be made available when the study is complete?

Reviewer #1: Yes

5. Is the manuscript presented in an intelligible fashion and written in standard English?

Reviewer #1: Yes

6. Review Comments to the Author

You may also provide optional suggestions and comments to authors that they might find helpful in planning their study.

Reviewer #1: No further comments to add to the paper. My review of this protocol is complete and I have no additional comments.

7. PLOS authors have the option to publish the peer review history of their article (what does this mean?). If published, this will include your full peer review and any attached files.

Reviewer #1: No

---

## [Editor Report · Acceptance letter]

25 Jan 2021

PONE-D-20-10082R2 

Evaluating Equality in Prescribing Novel Oral Anticoagulants (NOACs) in England: The Protocol of a Bayesian Small Area Analysis 

Dear Dr. Ahmadvand:

I'm pleased to inform you that your manuscript has been deemed suitable for publication in PLOS ONE. Congratulations! Your manuscript is now with our production department. 

Kind regards, 

on behalf of

Dr. Michele Tizzoni 

Academic Editor

PLOS ONE